# DISTRIBUTION EMBEDDING NETWORK FOR META-LEARNING WITH VARIABLE-LENGTH INPUT

## ABSTRACT

We propose Distribution Embedding Network (DEN) for meta-learning, which is designed for applications where both the data distribution and the *number* of features could vary across tasks. DEN first transforms features using a learned piecewise linear function, then learns an embedding of the transformed data distribution, and finally classifies examples based on the distribution embedding. We show that the parameters of the distribution embedding and the classification modules can be shared across tasks. We propose a novel methodology to mass-simulate binary classification training tasks, and demonstrate that DEN outperforms existing methods in a number of test tasks in numerical studies.

## 1 INTRODUCTION

Deep learning has made substantial progress in a variety of tasks in image classification (e.g., He et al., 2016), object detection (e.g., Redmon & Farhadi, 2017; He et al., 2017), machine translation (e.g., Vaswani et al., 2017) and natural language understanding (e.g., Devlin et al., 2019). These achievements rely on efficient gradient-based optimization algorithms (e.g., Duchi et al., 2011; Sutskever et al., 2013; Kingma & Ba, 2015) as well as a large number of labeled examples to train highly flexible deep learning models. However, in many applications, it is prohibitively expensive or impossible to collect a large amount of labeled training data, calling for techniques that can learn from small labeled data. Meta-learning aims to tackle the small data problem by training a model on labeled data from a number of related tasks, with the goal to learn a model that can perform well on similar, but unseen future tasks with only a small amount of labeled training data.

In this work, we propose a meta-learning model for classification using Distribution Embedding Networks (DEN). Unlike many existing meta-learning algorithms that assume a fixed feature set across tasks, DEN is designed for applications where both the distribution of features and the *number* of features could vary across tasks. For example, we may use DEN to learn the optimal aggregator of an ensemble of models to replace the naive majority vote, where in different aggregation tasks, the distribution of model outputs and the number of models in the ensemble could be different.

On a high level, DEN first applies a learned feature transformation that allows features to be transformed into the same distribution family across tasks. It then uses a neural network to learn an embedding of the transformed data distribution. Finally, given the learned distribution embedding, together with the transformed features, DEN classifies examples using a Deep Sets architecture (Zaheer et al., 2017), enabling it to be applied to variable-length inputs. To adapt the model on a new task, we only update the feature transformations with relatively few parameters.

## 2 RELATED WORK

There are multiple generic techniques applied to the meta-learning problem in the literature. The first camp of approaches learn similarities between pairs of examples. When presented with a new task with a small set of labeled examples, these methods classify unlabeled data based on their similarities with labeled ones. These methods include Matching Net (Vinyals et al., 2016) and Prototypical Net (Snell et al., 2017), which learn a distance metric between examples. Siamese Net (Koch et al., 2015) and Relation Net (Sung et al., 2018) use twin towers to learn the relationship between examples. Learn Net (Bertinetto et al., 2016) proposes having class specific weights for

the towers. Satorras & Estrach (2018) learns the similarity metric using a Graph Neural Network, and Transductive Propagation Network (Liu et al., 2019) classifies all unlabeled data at once by exploring the manifold structure of the new class space.

The second camp of optimization-based meta-learning aims to find a good starting point model, so that when presented with a new task, the meta model can quickly adapt to perform well on the new task with a small number of gradient steps. MAML (Finn et al., 2017) designs a learning algorithm, such that the expected loss of the learned meta-model on new tasks after one gradient step is minimized. Meta-Learner LSTM (Ravi & Larochelle, 2017) modifies the classical gradient steps with learned gradient update weights, which are trained to minimize the validation loss. More recently, LEO (Rusu et al., 2019) extends MAML and utilizes Relation Net to learn a low-dimensional latent embedding of model parameters and performs optimization-based meta-learning from this space.

Another camp of methods use internal or external memory for meta-learning. They include MANN (Santoro et al., 2016) and Meta Net (Munkhdalai & Yu, 2017), which store the past knowledge in external memory and internal model activations, respectively. New examples are classified by retrieving relevant information from the memory.

Our proposal does not take the above three routes. Rather, DEN is conceptually similar to topic modeling that learns the latent context variable. For example, Neural Statistician (Edwards & Storkey, 2017) considers the hierarchical generative process and uses variational autoencoder (Kingma & Welling, 2014) to learn the latent vector that summarizes the dataset in an unsupervised fashion. Similar proposals include variational homoencoder (Hewitt et al., 2018) and CNP (Garnelo et al., 2018). In comparison, DEN is a supervised procedure, which learns an embedding of the data distribution. We then utilize this distribution embedding for classification on unseen examples.

## 3 Notations

In this paper, we use bold upper case letters to denote matrices (e.g., $\boldsymbol{X}$), bold lower case letters to denote vectors (e.g., $\boldsymbol{x}$), italic lower case letters to denote scalars (e.g., $x$), normal text to denote random variables (e.g., x), and bold normal text to denote random vectors (e.g., $\mathbf{x}$).

Let $\mathbb{T}_1, \ldots, \mathbb{T}_M$ be $M$ training tasks, following some task distribution $P$. In training task $\mathbb{T}_i$, we observe a set of $n_i$ independent features and label pairs, $(\boldsymbol{X}_{\mathbb{T}_i}, \boldsymbol{y}_{\mathbb{T}_i})$, where $\boldsymbol{X}_{\mathbb{T}_i} = [\boldsymbol{x}_{\mathbb{T}_i}^1, \ldots, \boldsymbol{x}_{\mathbb{T}_i}^{d_i}] \in \mathbb{R}^{n_i \times d_i}$ is the $d_i$ dimensional real-valued feature matrix, $\boldsymbol{x}_{\mathbb{T}_i}^j \in \mathbb{R}^{n_i}$ is the $j$-th feature of task $\mathbb{T}_i$ and $\boldsymbol{y}_{\mathbb{T}_i} \in \{0, 1\}^{n_i}$ is the binary label vector. We assume that the label is binary for simplicity of presentation. Our proposed model can be trivially extended to multiclass classification problems. We use $P_{\mathbb{T}_i}$ to denote the joint distribution of $(\mathbf{x}_{\mathbb{T}_i}, \mathbf{y}_{\mathbb{T}_i})$.

We first train DEN on the training tasks $\{\mathbb{T}_1, \ldots, \mathbb{T}_M\}$. Given a new task $\mathbb{S}$ with a small set of labeled examples, we fine-tune the trained model on $\mathbb{S}$ using these labeled examples. The final model is then applied to unlabeled examples in $\mathbb{S}$ for classification. The set of labeled examples is called the *support set* and the set of unlabeled examples is called the *query set*.

## 4 Distribution Embedding Network

To motivate our proposal, we first consider the problem of minimizing the risk on a single task $\mathbb{T}$:

$$\hat{\boldsymbol{\theta}}_{\mathbb{T}} = \underset{\boldsymbol{\theta} \in \Theta}{\arg \min} \, \mathbb{E}_{(\mathbf{x}_{\mathbb{T}}, \mathbf{y}_{\mathbb{T}}) \sim P_{\mathbb{T}}} \left[ L(f(\mathbf{x}_{\mathbb{T}}; \boldsymbol{\theta}), \mathbf{y}_{\mathbb{T}}) \right], \tag{1}$$

where $f(\cdot; \boldsymbol{\theta})$ is a model with parameter $\boldsymbol{\theta}$ and $L$ is the loss function.

**Lemma 1.** *Assume the joint distribution $P_{\mathbb{T}}$ has a probability density (mass) function $q(\cdot; \boldsymbol{\eta}_{\mathbb{T}})$. Then the optimizer $\hat{\boldsymbol{\theta}}_{\mathbb{T}}$ is of the form $\phi_{L,f,q}^*(\boldsymbol{\eta}_{\mathbb{T}})$, where $\phi_{L,f,q}^*$ is some deterministic function depending on the loss $L$, the model $f$ and the density $q$.*

The proof of Lemma 1 can be found in Appendix A. It suggests that, when the joint distribution $P_{\mathbb{T}}$ is in a parametric family, the dependency of $\hat{\boldsymbol{\theta}}_{\mathbb{T}}$ on the task $\mathbb{T}$ is through two parts: the functional form of the density $q$ and the distribution parameter $\boldsymbol{\eta}_{\mathbb{T}}$.

Now, if the joint distributions of all training tasks $\{\mathbb{T}_1, \ldots, \mathbb{T}_M\}$ are in the same parametric family, then there exists a common function $\phi^*_{L,f,q}$ such that $\hat{\boldsymbol{\theta}}_{\mathbb{T}_i} = \phi^*_{L,f,q}(\boldsymbol{\eta}_{\mathbb{T}_i})$ for every $i \in \{1, \ldots, M\}$. Hence, we may reparameterize $f(\boldsymbol{x}_\mathbb{T}; \phi^*_{L,f,q}(\boldsymbol{\eta}_\mathbb{T}))$ as $f'(\boldsymbol{x}_\mathbb{T}, \boldsymbol{\eta}_\mathbb{T}; \boldsymbol{\gamma})$, and learn the new model parameter $\boldsymbol{\gamma}$ from observations $(\boldsymbol{X}_{\mathbb{T}_i}, \boldsymbol{y}_{\mathbb{T}_i})$ and some estimate $\hat{\boldsymbol{\eta}}_{\mathbb{T}_i}$. One advantage of this reparameterization is that the optimal choice of $\boldsymbol{\gamma}$ is now task-independent, so, once learned from training tasks, it can be transferred to new unseen tasks whose data distribution falls in the same parametric family. Moreover, since the data distributions of all tasks are in the same parametric family, we can learn a shared model to estimate $\hat{\boldsymbol{\eta}}_\mathbb{T}$ for all $\mathbb{T}$.

In practice, the data distribution could vary greatly across tasks. Hence, for each task $\mathbb{T}$, we propose to apply a transformation $c_\mathbb{T}$ to the original features and work with $(c_\mathbb{T}(\mathbf{x}_\mathbb{T}), \mathbf{y}_\mathbb{T})$, allowing it to unify the distribution family of $(c_\mathbb{T}(\mathbf{x}_\mathbb{T}), \mathbf{y}_\mathbb{T})$.

To summarize, our proposal, DEN, consists of three building blocks. Firstly, we apply a transformation layer $c_\mathbb{T}$ to the original features. Secondly, we use a distribution embedding module to obtain a distribution embedding $\boldsymbol{s}_\mathbb{T}$ of the transformed data $(c_\mathbb{T}(\mathbf{x}_\mathbb{T}), \mathbf{y}_\mathbb{T})$. This corresponds to the distribution parameter $\boldsymbol{\eta}_\mathbb{T}$ discussed above. Finally, we use a classification module to output a prediction based on the transformed feature $c_\mathbb{T}(\boldsymbol{x}_\mathbb{T})$ and the distribution embedding $\boldsymbol{s}_\mathbb{T}$. Here the last two modules are shared across tasks.

## 4.1 Learning the Distribution Embedding

To illustrate the idea of learning a compact distribution embedding, we first consider a special case that $\mathbf{x}_\mathbb{T}|(\mathbf{y}_\mathbb{T} = k) \sim \mathcal{N}_d(\boldsymbol{\mu}_{\mathbb{T},k}, \boldsymbol{\Sigma}_{\mathbb{T},k})$ for $k \in \{0, 1\}$. With the normality assumption, $P_\mathbb{T}$ is uniquely characterized by its first moment vectors $\mathbb{E}[\mathbf{x}_\mathbb{T}|(\mathbf{y}_\mathbb{T} = k)]$, second moment matrices $\mathbb{E}[\mathbf{x}_\mathbb{T}\mathbf{x}_\mathbb{T}^\top|(\mathbf{y}_\mathbb{T} = k)]$, and the label positive rate $\mathbb{P}(\mathbf{y}_\mathbb{T} = 1)$. Given an i.i.d. sample $(\boldsymbol{X}_\mathbb{T}, \boldsymbol{y}_\mathbb{T})$ where $\boldsymbol{X}_\mathbb{T} := [\boldsymbol{x}_\mathbb{T}^1, \ldots, \boldsymbol{x}_\mathbb{T}^d] \in \mathbb{R}^{n \times d}$ and $\boldsymbol{y}_\mathbb{T} \in \{0, 1\}^n$, we may estimate these quantities by sample moments, leading to a compact distribution embedding

$$\boldsymbol{s}_\mathbb{T} = \Big[ \overline{\boldsymbol{x}^1_{\mathbb{T},0}}, \ldots, \overline{\boldsymbol{x}^d_{\mathbb{T},0}}, \overline{\boldsymbol{x}^1_{\mathbb{T},0} \odot \boldsymbol{x}^1_{\mathbb{T},0}}, \ldots, \overline{\boldsymbol{x}^1_{\mathbb{T},0} \odot \boldsymbol{x}^d_{\mathbb{T},0}}, \ldots, \overline{\boldsymbol{x}^d_{\mathbb{T},0} \odot \boldsymbol{x}^d_{\mathbb{T},0}}, $$
$$\overline{\boldsymbol{x}^1_{\mathbb{T},1}}, \ldots, \overline{\boldsymbol{x}^d_{\mathbb{T},1}}, \overline{\boldsymbol{x}^1_{\mathbb{T},1} \odot \boldsymbol{x}^1_{\mathbb{T},1}}, \ldots, \overline{\boldsymbol{x}^1_{\mathbb{T},1} \odot \boldsymbol{x}^d_{\mathbb{T},1}}, \ldots, \overline{\boldsymbol{x}^d_{\mathbb{T},1} \odot \boldsymbol{x}^d_{\mathbb{T},1}}, \overline{\boldsymbol{y}_\mathbb{T}} \Big], \qquad (2)$$

where $\boldsymbol{x}^j_{\mathbb{T},k}$ denotes the $j$-th feature vector of examples with label $k$, $\overline{\boldsymbol{x}}$ denotes the arithmetic average of its elements for a vector $\boldsymbol{x}$, and $\odot$ is the element-wise product.

There are two notable characteristics in this distribution embedding. Firstly, every element in $\boldsymbol{s}_\mathbb{T}$ is of an average form over examples within the same class. This motivates us to use a batch average layer in DEN when obtaining the distribution embedding. Secondly, the embedding $\boldsymbol{s}_\mathbb{T}$ only involves quantities of single features and pairs of features. As discussed in Section 4.1.2, this allows us to decompose the whole distribution embedding vector of task-dependent dimension (e.g., $2d^2 + 2d + 1$ for Gaussian data) into sub-vectors of fixed dimension, opening up an opportunity to handle variable-length features using a Deep Sets architecture (Zaheer et al., 2017).

### 4.1.1 Non-Gaussian Features

In practice, the normality assumption is usually overly simplified. To address this issue, one simple strategy is to first apply a transformation $c_\mathbb{T} : \mathbb{R}^d \to \mathbb{R}^d$ on each task so that $c_\mathbb{T}(\mathbf{x}_\mathbb{T})|(\mathbf{y}_\mathbb{T} = k)$ follows a multivariate normal distribution. However, such transformations are not guaranteed to exist. It is also challenging to analytically construct such transformations, even if they exist.

Instead of analytically constructing the transformations $c_\mathbb{T}$, we propose to *learn* a piecewise linear function (PLF) $c_\mathbb{T}^j : \mathbb{R} \to \mathbb{R}$ for each feature $j$ and task $\mathbb{T}$, i.e.,

$$z_\mathbb{T}^j := c_\mathbb{T}^j(x; \boldsymbol{k}_\mathbb{T}^j, \boldsymbol{\alpha}_\mathbb{T}^j) = \sum_{i=1}^{K-1} \left( \alpha^j_{\mathbb{T},i} + \frac{x - k^j_{\mathbb{T},i}}{k^j_{\mathbb{T},i+1} - k^j_{\mathbb{T},i}} \left( \alpha^j_{\mathbb{T},i+1} - \alpha^j_{\mathbb{T},i} \right) \right) \mathbb{1}\left( k^j_{\mathbb{T},i} \le x \le k^j_{\mathbb{T},i+1} \right),$$
$$(3)$$

where $\boldsymbol{k}_{\mathbb{T}}^{j} := [k_{\mathbb{T},1}^{j}, \ldots, k_{\mathbb{T},K}^{j}] \in \mathbb{R}^{K}$, $k_{\mathbb{T},1}^{j} < k_{\mathbb{T},2}^{j} < \cdots < k_{\mathbb{T},K}^{j}$, is the vector of *predetermined* keypoints that spans the domain[1] of $x_{\mathbb{T}}^{j}$, and $\boldsymbol{\alpha}_{\mathbb{T}}^{j} := [\alpha_{\mathbb{T},1}^{j}, \ldots, \alpha_{\mathbb{T},K}^{j}] \in \mathbb{R}^{K}$ is the parameter vector of the PLF, characterizing its output at each keypoint. The PLF can be optionally constrained to be monotonic with $\alpha_{\mathbb{T},i}^{j} \leq \alpha_{\mathbb{T},i+1}^{j}$ for all $i = 1, \ldots, K-1$, which serves as a regularization and can be satisfied through parameter projections during training through, e.g., projected SGD.

PLFs can implement compact one-dimensional non-linearities that can be learned with a small sample. They are universal approximators in this space: with enough keypoints, they can approximate any bounded continuous function. Although it is not guaranteed that features transformed by PLFs satisfy the normality assumption, PLFs make it possible that transformed features (approximately) belong to the *same parametric family* for all tasks.

Since this parametric family is not necessarily normal, the sample moments used in (2) are no longer appropriate. Fortunately, the next lemma indicates that, as long as this parametric family is an exponential family, the distribution embedding can be chosen as an average form.

**Lemma 2.** *Let $Q$ be a probability distribution in an exponential family with density $q(\boldsymbol{u}; \boldsymbol{\eta}) := B(\boldsymbol{u}) \exp[\lambda(\boldsymbol{\eta}) \cdot S(\boldsymbol{u}) - A(\boldsymbol{\eta})]$. Let $\boldsymbol{u}_1, \ldots, \boldsymbol{u}_n \overset{i.i.d.}{\sim} Q$. Then $\sum_{i=1}^{n} S(\boldsymbol{u}_i)$ is a sufficient statistic for $\boldsymbol{\eta}$.*

According to Lemma 2, if the conditional distribution $P_{\mathbb{T}}(\mathbf{z}_{\mathbb{T}} | \mathrm{y}_{\mathbb{T}} = k)$ belongs to the same exponential family for all tasks $\mathbb{T}$ and $k \in \{0, 1\}$, then there exists a task-independent function $S$ such that $\overline{S(\boldsymbol{z}_{\mathbb{T},k})}$ is sufficient for $P_{\mathbb{T}}(\mathbf{z}_{\mathbb{T}} | \mathrm{y}_{\mathbb{T}} = k)$. Consequently, we may use $\boldsymbol{s}_{\mathbb{T}} := [\overline{S(\boldsymbol{z}_{\mathbb{T},0})}, \overline{S(\boldsymbol{z}_{\mathbb{T},1})}, \overline{\boldsymbol{y}_{\mathbb{T}}}]$ as a distribution embedding of the joint probability $P_{\mathbb{T}}$, where $S$ can be encoded as a neural network.

### 4.1.2 VARIABLE-LENGTH FEATURES

To handle variable-length features, instead of learning the whole distribution embedding $\boldsymbol{s}_{\mathbb{T}}$ directly, we decompose the joint distribution of $(\mathbf{z}_{\mathbb{T}}, \mathrm{y}_{\mathbb{T}})$ into smaller pieces: conditional probabilities $z_{\mathbb{T}}^{i_1}, \ldots, z_{\mathbb{T}}^{i_r} | \mathrm{y}$ for all *r-subsets* $\{i_1, \ldots, i_r\} \subset \{1, \ldots, d\}$ and the marginal $\mathrm{y}$, where $r$ is a hyperparameter shared across tasks. The optimal choice of $r$ should depend on the distribution of $(\mathbf{z}_{\mathbb{T}}, \mathrm{y}_{\mathbb{T}})$. For example, as discussed in (2), when $\mathbf{z}_{\mathbb{T}}$ follows a multivariate normal distribution conditional on $\mathrm{y}_{\mathbb{T}}$, $r = 2$ can sufficiently characterize the joint distribution of $(\mathbf{z}_{\mathbb{T}}, \mathrm{y}_{\mathbb{T}})$. We will use $r = 2$ in all numerical studies in Section 5.

Following the discussion of Lemma 2 in Section 4.1.1, we use a learnable model $g$, shared across tasks, to derive a distribution embedding vector for each subset $\{i_1, \ldots, i_r\} \subset \{1, \ldots, d\}$:

$$\boldsymbol{s}_{\mathbb{T}}^{i_1, \ldots, i_r} := \left[ \overline{g\left(\left[\boldsymbol{z}_{\mathbb{T},0}^{i_1}, \ldots, \boldsymbol{z}_{\mathbb{T},0}^{i_r}\right]\right)}, \overline{g\left(\left[\boldsymbol{z}_{\mathbb{T},1}^{i_1}, \ldots, \boldsymbol{z}_{\mathbb{T},1}^{i_r}\right]\right)}, \overline{\boldsymbol{y}_{\mathbb{T}}} \right], \tag{4}$$

where the average is taken with respect to a training batch during training or the support set during testing. We will discuss it in more detail in Section 4.3.

An alternative formulation of $\boldsymbol{s}_{\mathbb{T}}$ is to directly embed the joint distribution of $(z_{\mathbb{T}}^{i_1}, \ldots, z_{\mathbb{T}}^{i_r}, \mathrm{y})$,

$$\boldsymbol{s}_{\mathbb{T}}^{i_1, \ldots, i_r} = \overline{\tilde{g}\left(\left[z_{\mathbb{T}}^{i_1}, \ldots, z_{\mathbb{T}}^{i_r}, y_{\mathbb{T}}\right]\right)}. \tag{5}$$

Both formulations in (4) and (5) work well in numerical studies presented in Section 5.

### 4.2 CLASSIFICATION WITH DISTRIBUTION EMBEDDING

With a distribution embedding $\boldsymbol{s}_{\mathbb{T}}^{i_1, \ldots, i_r}$ for each subset $\{i_1, \ldots, i_r\}$, we group the embedding $\boldsymbol{s}_{\mathbb{T}}^{i_1, \ldots, i_r}$ with its associated transformed features $(z_{\mathbb{T}}^{i_1}, \ldots, z_{\mathbb{T}}^{i_r})$, and then use a Deep Sets structure (Zaheer et al., 2017) over the set of all $M$-subsets of features:

$$\hat{y} = \psi \left( \binom{d}{r}^{-1} \sum_{\{i_1, \ldots, i_r\} \subset \{1, \ldots, d\}} h\left(\left[z_{\mathbb{T}}^{i_1}, \ldots, z_{\mathbb{T}}^{i_r}, \boldsymbol{s}_{\mathbb{T}}^{i_1, \ldots, i_r}\right]\right) \right), \tag{6}$$

[1]For instance, they can be chosen as quantiles of the distribution of $x_{\mathbb{T}}^{j}$.

which is able to accommodate a variable number of $h$'s, and in turn accommodate a variable number of inputs. Based on Theorem 7 in Zaheer et al. (2017), (6) could approximate any permutation invariant continuous function in $h$. Note that $s_{\mathbb{T}}^{i_1,\ldots,i_r}$ is the *average embedding*, and all examples within the same batch/task share the same $s_{\mathbb{T}}^{i_1,\ldots,i_r}$ (see details in Section 4.3).

### 4.3 TRAINING AND INFERENCE

Figure 1 shows a high level summary of our model graph during training, where $\mathbb{T}_1,\ldots,\mathbb{T}_m$ are training tasks. In each gradient step, we first randomly sample a task $\mathbb{T}_i \in \{\mathbb{T}_1,\ldots,\mathbb{T}_m\}$. We then sample two disjoint batches $A$ and $B$ of training features and labels from $(\boldsymbol{X}_{\mathbb{T}_i}, \boldsymbol{y}_{\mathbb{T}_i})$. The two batches of features are first transformed using PLFs in (3). We then use (4) or (5) to obtain a distribution embedding, taking the average with respect to the sample in batch B. Next, we use the average distribution embedding to make predictions on batch $A$ using the Deep Sets formulation in (6). The use of two batches is to reflect the scenario that during inference, we use the support set (i.e., batch $B$) to obtain the distribution embedding of the task, with which we classify query set examples (i.e., batch $A$). This procedure is similar to the idea of episode-based training introduced in Vinyals et al. (2016). Note that during training, $s_{\mathbb{T}_i}$ is identical across examples within the same batch, but it could vary (even within the same task) across batches.

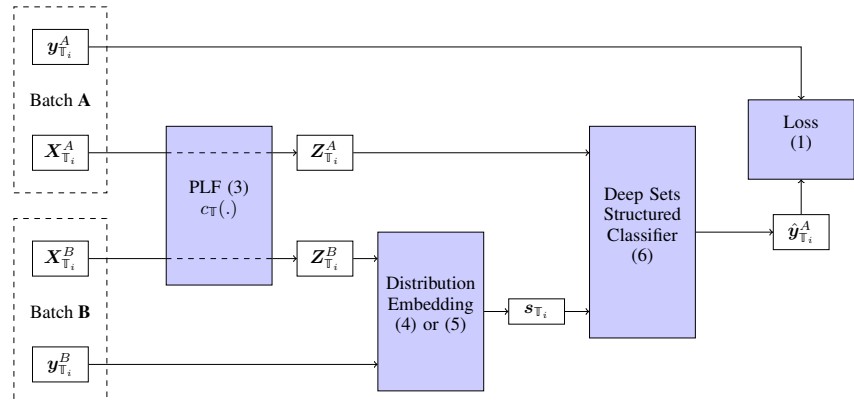

Figure 1: Block diagram of the model graph during training. We resample batches A and B and Task $\mathbb{T}_i$ after each gradient step.

If PLFs are able to approximately transform the features into the same parametric family, then the rest of the network can be task-agnostic. Thus, after training on $\mathbb{T}_1,\ldots,\mathbb{T}_m$, for each new task $\mathbb{S}$, we learn a set of new PLFs (3), while keeping the weights of other layers fixed. Because PLFs only have a small number of parameters, they can be trained on a small support set $(\boldsymbol{X}_{\mathbb{S}}, \boldsymbol{y}_{\mathbb{S}})$.

During inference, we first use learned PLFs in (3) to transform features in both the support set and the query set. We then utilize the learned distribution embedding module to obtain $s_{\mathbb{S}}$, where the average in (5) and (4) is taken over the whole support set. Finally, the embedding $s_{\mathbb{S}}$ and PLF-transformed query set features are used to classify query set examples using (6).

## 5 NUMERICAL STUDIES

In this section, we numerically compare the two formulations of DEN, Conditional DEN (4) and Joint DEN (5) (with $r = 2$), with a range of baseline methods, including Prototypical Net (Snell et al., 2017), Relation Net (Sung et al., 2018) and MAML (Finn et al., 2017) applied to a ReLU-activated deep neural network (DNN). All hyperparameters are chosen based on cross-validation on training tasks, and are summarized in Appendix B. We also compare our method against DNNs trained directly on the support set (Direct DNN), whose hyperparameters were chosen based on cross-validation on the support set.[2]

---

[2]We will make the code of the experiments of Section 5.1 publicly available upon acceptance of this paper.

## 5.1 GENERATE TRAINING TASKS THROUGH CONTROLLED SIMULATION

To generate training tasks, we adopt a novel approach. We take seven multiclass image classification datasets: CIFAR-10, CIFAR-100 (Krizhevsky, 2009), MNIST (LeCun et al., 2010), Fashion MNIST (Xiao et al., 2017), EMNIST (Cohen et al., 2017), Kuzushiji MNIST (KMNIST; Clanuwat et al., 2018) and Street View House Numbers (SVHN; Netzer et al., 2011). On each dataset, we pick nine equally spaced cutoffs and binarize the labels based on whether the class id is below the cutoff or not. This gives rise to nine binary classification tasks for each dataset with positive label proportion in $\{0.1, 0.2, \ldots, 0.9\}$. In summary, we collect $7 \times 9 = 63$ binary classification tasks.

To generate features for DEN, we build 50 image classifiers on each task $\mathbb{T}_i \in \{\mathbb{T}_1, \ldots, \mathbb{T}_{63}\}$, and take classification scores on the test set as features $\boldsymbol{x}_{\mathbb{T}_i} \in (0, 1)^{50}$. The 50 convolutional image classifiers all have the structure $\texttt{Conv}(f, k) \rightarrow \texttt{MaxPool}(p) \rightarrow \texttt{Conv}(f, k) \rightarrow \texttt{MaxPool}(p) \rightarrow \cdots \rightarrow \texttt{Conv}(f, k) \rightarrow \texttt{Dense}(u) \rightarrow \texttt{Dense}(u) \rightarrow \cdots \rightarrow \texttt{Dense}(1)$, where the number of dense layers $d \in [1, 4]$, the number of convoluted layers $c \in [0, 3]$, filters and units $f, u \in [2, 511]$, kernel and pool sizes $k, p \in [2, 5]$, and training epochs $e \in [1, 24]$ are uniformly sampled. Note that these 50 classifiers range from linear classifiers to ReLU-activated deep convolutional neural networks, with accuracy ranging from below 0.6 to over 0.99.

Finally, to augment the training data, we apply sub-sampling during training. In each training step, after selecting a task and two disjoint batches of training examples as discussed in Section 4.3, we randomly pick $C < 50$ classifiers to construct a sub-task that aims to aggregate the classification scores of $C$ classifiers. Here $C$ could vary across training steps, and thus the joint distribution $P_{\mathbb{T}}$ would be different in different training steps, helping DEN learn from a diverse set of $P_{\mathbb{T}}$. By increasing the number of datasets, the number of label cutoffs and the number of classifiers, we could effectively generate a large number of training (sub-)tasks and examples for DEN.

## 5.2 META-LEARNING ON MODEL AGGREGATION

In this section, we study the performance of DEN in aggregating the outputs of an ensemble of classifiers. We also study the impact of fine-tuning PLF on the performance of DEN. As discussed in Section 4, fine-tuning PLF is a powerful way to make DEN adapt to a wide variety of tasks with different input distributions. However, in the scenario when the distributions of training and target tasks are similar, fine-tuning PLF can introduce overfitting that outweighs its benefit, especially when the support set is small compared to the number of features.

To introduce similar training and target tasks, we use $5 \times 9 = 45$ tasks derived from CIFAR-10, CIFAR-100, MNIST, Fashion MNIST and EMNIST to train DEN and other meta-learning methods. The model architectures and hyperparameters are presented in Appendix B. Note that because the distributions of classifier outputs are similar, we learn a single set of $C$ PLFs during training across tasks for simplicity. We then pick four test tasks from SVHN and KMNIST (two from each dataset) of different difficulties. The average AUC across 50 classifiers are 68.28%, 78.11%, 91.51%, and 87.58%, respectively. Given a test task, we randomly select 100 sets of $C$ classifiers among the 50 candidate classifiers, which results in 100 aggregation sub-tasks for each of the four test tasks. For each aggregation sub-tasks, we form a support set with 50 labeled examples and a disjoint query set with 8000 examples. "Direct DNN" is trained directly on the 50 support set examples. For DNN with MAML, we fine-tune the last DNN layer for 5 epochs on the support set. For DEN, we either fine-tune the first layer of $C$ PLFs for 10 epochs, or take $C$ PLFs directly from the last training epoch without fine-tuning. The entire training and fine-tuning process is repeated 5 times. For each test task, we report the average AUC score across $5 \times 100$ aggregation sub-tasks and its estimated standard error.

In Table 1, we present the result where we aggregate $C = 25$ classifiers. Average and product rules are simple heuristics that make predictions based on the average and product of the 25 classification outputs, respectively. In Table 2, we show the result where the number of classifiers $C$ to be aggregated is sampled uniformly from $[13, 25]$ across those 500 trials. To allow baseline methods to take varying number of features, we repeat and add a random subset of $(25 - C)$ features so that all inputs have 25 features. We observe that DEN significantly outperforms other methods in seven out of eight tasks.

Table 1: Test % AUC (standard error) when aggregating 25 classifiers. Bold fonts indicate the best method at 95% significance level.

| Method | Test AUC (%) | | | |
|---|---|---|---|---|
| | Task 1 | Task 2 | Task 3 | Task 4 |
| Average rule | 86.68 (0.176) | 88.80 (0.048) | 97.54 (0.030) | 96.27 (0.099) |
| Product rule | 87.74 (0.168) | 88.96 (0.046) | **98.37** (0.009) | 97.37 (0.044) |
| Direct DNN | 85.95 (0.362) | 89.53 (0.032) | 97.88 (0.048) | 97.56 (0.033) |
| Prototypical | 91.06 (0.051) | 89.95 (0.029) | 98.19 (0.006) | 97.51 (0.017) |
| Relation | 83.26 (0.310) | 88.71 (0.025) | 97.18 (0.023) | 95.64 (0.079) |
| DNN + MAML | 86.05 (0.161) | 88.74 (0.033) | 97.42 (0.022) | 96.02 (0.068) |
| Joint DEN w/o Fine-Tuining | 91.77 (0.030) | **90.26** (0.021) | 98.15 (0.006) | **98.43** (0.003) |
| Joint DEN w Fine-Tuining | **91.87** (0.029) | 90.04 (0.020) | 97.55 (0.007) | 97.51 (0.006) |
| Cond. DEN w/o Fine-Tuining | 91.76 (0.032) | 90.20 (0.021) | 98.18 (0.005) | 98.41 (0.003) |
| Cond. DEN w Fine-Tuining | 91.80 (0.030) | 89.77 (0.021) | 97.38 (0.007) | 97.23 (0.006) |

Table 2: Test % AUC (standard error) when aggregating variable number of classifiers. Bold fonts indicate the best method at 95% significance level.

| Method | Test AUC (%) | | | |
|---|---|---|---|---|
| | Task 1 | Task 2 | Task 3 | Task 4 |
| Average rule | 86.05 (0.437) | 88.65 (0.080) | 97.43 (0.059) | 96.09 (0.139) |
| Product rule | 87.06 (0.462) | 88.80 (0.072) | **98.34** (0.017) | 97.27 (0.059) |
| Direct DNN | 83.83 (0.595) | 89.11 (0.057) | 97.71 (0.050) | 97.23 (0.107) |
| Prototypical | 90.02 (0.107) | 89.65 (0.041) | 97.94 (0.015) | 97.57 (0.016) |
| Relation | 80.85 (0.646) | 88.53 (0.038) | 97.11 (0.039) | 95.03 (0.123) |
| DNN + MAML | 85.29 (0.229) | 88.51 (0.044) | 97.32 (0.036) | 95.88 (0.085) |
| Joint DEN w/o Fine-Tuning | **91.21** (0.098) | **90.07** (0.029) | **98.31** (0.007) | **98.18** (0.006) |
| Joint DEN w Fine-Tuning | **91.34** (0.030) | 89.74 (0.022) | 97.29 (0.030) | 97.10 (0.004) |
| Cond. DEN w/o Fine-Tuning | **91.17** (0.092) | 89.95 (0.028) | 97.95 (0.008) | 98.10 (0.006) |
| Cond. DEN w Fine-Tuning | **91.29** (0.030) | 89.83 (0.022) | 97.60 (0.025) | 97.47 (0.004) |

Results in Table 1 and 2 also show that in 15 out of 16 comparisons, DEN without fine-tuning is statistically no worse than DEN with fine-tuning on the PLF layer. This may suggest that fine-tuning the PLF layer is not necessary when the input distribution is similar among tasks.

## 5.3 Meta-Learning on Real Datasets

Finally, we apply DEN to seven target tasks from three real datasets: Diamonds, Nomao and Puzzles. We give a short description of each dataset below, and list the features in Appendix D.

- With Diamonds data[3], we use five features to classify whether the price of the diamond is above $5325, $2402, or $951. This results in three binary classification tasks with positive label proportions of 25%, 50%, and 75%, respectively.

- With Nomao data[4], we use seven features to classify whether two business entities are identical. Positive examples account for 71% of the data.

- With Puzzles data[5], we use six features to classify whether the number of units sold for each puzzle in a six months period is above 93, 45, or 24. The support set (with 155 puzzles) covers the first six months period, whereas the query set (with 367 puzzles) covers the second and third six months periods. The three cutoffs result in positive label proportions of 25%, 49% and 74% in the support set, and 23%, 51% and 73% in the query set.

---

[3]https://www.kaggle.com/shivam2503/diamonds
[4]https://archive.ics.uci.edu/ml/datasets/Nomao
[5]https://www.kaggle.com/dbahri/puzzles

In this study, we use all 63 tasks described in Section 5.1 during training. We train a single Joint DEN and a single Conditional Den model for all seven tasks with $C \leq 9$, and fine-tune the PLF on 50 support set examples for Diamonds and Nomao tasks, and on 155 support set examples for Puzzles tasks. For other methods, we train a different model for each dataset (three in total), using a suitable number of features. In the Nomao data, six out of seven features contain missing values. For DEN, we learn a missing value embedding via the PLF, whereas for other methods, we add six missing value indicator features, each corresponding to a feature with missing value.

We repeat the whole procedure for 20 times with different random seeds in initialization and selection of training batches and report the average AUC and standard error in Tables 3, 4 and 5. DEN significantly outperforms other methods in six out of seven tasks. Compared with other methods, DEN is especially impressive in the Nomao task, in which all features are monotonic, and six of them include missing values. Direct DNN performs well in two of the Puzzles tasks with 155 support set examples, indicating that the benefit of DEN diminishes with a large support set. Note that fine-tuning the PLF greatly improved the performance of DEN in these seven tasks, which shows that fine-tuning the PLF is helpful when the training and target tasks have different distributions.

Table 3: Test % AUC (standard error) on Diamonds data. We binarize price based on $951, $2402 and $5325 to generate three tasks. Bold fonts indicate the best method at 95% significance level.

| Method | Price > $951 | Price > $2402 | Price > $5325 |
|---|---|---|---|
| Direct DNN | 65.82 (3.077) | 90.62 (0.962) | 65.37 (2.971) |
| Prototypical | 82.10 (0.177) | 91.49 (0.195) | 92.98 (0.182) |
| Relation | 81.09 (1.568) | 88.75 (1.605) | 88.98 (1.177) |
| DNN + MAML | 79.30 (1.315) | 82.83 (1.541) | 90.41 (0.970) |
| Joint DEN | **99.58** (0.065) | 99.49 (0.048) | 99.24 (0.089) |
| Conditional DEN | 97.90 (0.166) | **99.67** (0.038) | **99.47** (0.057) |

Table 4: Test % AUC (standard error) on Nomao data. Bold fonts indicate the best method at 95% significance level.

| Direct DNN | Prototypical | Relation | DNN + MAML | Joint DEN | Cond. DEN |
|---|---|---|---|---|---|
| 69.60 (2.372) | 80.56 (0.559) | 52.32 (1.614) | 78.92 (2.224) | **95.61** (0.071) | 95.21 (0.096) |

Table 5: Test % AUC (standard error) on Puzzles data. We binarize number of units sold based on 24, 45 and 93 to generate three tasks. Bold fonts indicate the best method at 95% significance level.

| Method | Sales > 24 | Sales > 45 | Sales > 93 |
|---|---|---|---|
| Direct DNN | 64.37 (0.827) | **78.44** (0.463) | **79.83** (0.996) |
| Prototypical | 69.22 (0.220) | 73.77 (0.366) | **81.39** (0.582) |
| Relation | 62.40 (1.368) | 63.73 (1.792) | 63.83 (2.065) |
| DNN + MAML | 53.84 (0.519) | 54.92 (0.535) | 56.57 (0.699) |
| Joint DEN | 64.55 (0.844) | 75.68 (0.600) | 73.96 (0.279) |
| Conditional DEN | **72.62** (0.512) | **78.11** (0.533) | 75.50 (0.240) |

## 6  CONCLUSION

In this paper, we presented a novel meta-learning algorithm that can be applied to settings where both the distribution and number of input features vary across tasks. Most other meta-learning techniques do not readily handle such settings. In numerical studies, we demonstrated that an application of the proposed method on binary classification problems outperforms several meta-learning baselines.

The permutation invariance of the proposed structure is an interesting venue to explore in future work; see (Bloem-Reddy & Teh, 2020) for a thorough discussion on this topic. Furthermore, we can

use the Set Transformer (Lee et al., 2019) in place of the Deep Sets structure in the classifier block of DEN, and we leave the comparison to future work.

Our proposed method can be extended to multi-class classifications by revising the formulations in (4) and (5). DEN can also be combined with optimization based meta-learning methods, e.g., MAML, to further improve its performance. Finally, although convenient and intuitive, the one-dimensional PLFs are not able to account for the associations between features. Improving the first layer of DEN may lead to further performance improvement.

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

## A  APPENDIX: PROOF OF LEMMAS

*Proof of Lemma 1.* For simplicity, we assume the joint distribution $P_\mathbb{T}$ has probability density $q(\cdot; \boldsymbol{\eta})$. Define a map $\mathcal{L} : \Theta \times \mathbb{R}^r \to \mathbb{R}_+$ by

$$\mathcal{L}(\boldsymbol{\theta}, \boldsymbol{\eta}) := \mathbb{E}_{(\mathbf{x},\mathbf{y}) \sim P_\mathbb{T}}[L(f(\mathbf{x}; \boldsymbol{\theta}), \mathbf{y})] = \int L(f(\boldsymbol{x}; \boldsymbol{\theta}), y) q(\boldsymbol{x}, y; \boldsymbol{\eta}) d\boldsymbol{x} dy,$$

where $\mathcal{L}$ depends on the loss $L$, the model $f$ and the density $q$. Then the problem becomes

$$\hat{\boldsymbol{\theta}} = \underset{\boldsymbol{\theta} \in \Theta}{\arg\min} \, \mathcal{L}(\boldsymbol{\theta}, \boldsymbol{\eta}).$$

Hence, $\hat{\boldsymbol{\theta}}$ is of the form $\phi^*_{L,f,q}(\boldsymbol{\eta})$.  □

*Proof of Lemma 2.* By independence, the joint probability of $\{\boldsymbol{u}_i\}_{i=1}^n$ is

$$\prod_{i=1}^n q(\mathbf{u}_i; \boldsymbol{\eta}) = \prod_{i=1}^n B(\boldsymbol{u}_i) \exp\left[\lambda(\boldsymbol{\eta}) \cdot \sum_{i=1}^n S(\boldsymbol{u}_i) - nA(\boldsymbol{\eta})\right].$$

According to Fisher–Neyman factorization theorem (Halmos & Savage, 1949), the statistic $\boldsymbol{s}(\boldsymbol{U}) := \sum_{i=1}^n S(\boldsymbol{u}_i)$ is sufficient for $\boldsymbol{\eta}$.  □

## B  APPENDIX: MODEL STRUCTURES AND HYPERPARAMETERS

In this section, we report model structures and hyperparameters of all models we have considered in Section 5. We use cross-validation on $5 \times 9$ tasks derived from CIFAR-10, CIFAR-100, MNIST, Fashion MNIST and EMNIST to tune the hyperaprameters. Note that those 45 tasks were treated as training tasks in all experiments presented in Section 5.

For DEN, Prototypical Net and Relation Net, we trained each of them for 200 epochs using Adam optimizer with batch size 256 and the TensorFlow default learning rate, 0.001. Their model-related hyperparameters were chosen based on 5-fold cross-validation: in each fold, we trained the model on all but one training datasets and evaluated it on the remaining one[6], feeding with 500 supporting examples; we then selected the model with the highest AUC scores averaged over 5 folds. For DNN with MAML, after training it on the 36 tasks, we fine-tuned the last layer of the trained DNN on a support set of size 500 of each validation task, then evaluated it on the remaining examples of this validation task. For the DNN model directly trained on the support set, its hyperparameters were selected on the support set of each test task. So it could vary from task to task. We only report its model structure since there are many test tasks.

Based on cross-validation, we arrived at the following model architectures. We use the following architectures across all of our experiments. We use $m$-Dense($u$) to represent $m$ consecutive ReLU-activated dense layers with $u$ units. With a bit abuse of notation, we use Dense(1) to represent a sigmoid-activated dense layer with 1 unit. We denote by BatchAvg an average layer over examples in a batch, and denote by PairAvg an average layer over pairs of features.

- **Direct DNN**: $m$-Dense($u$) $\to$ Dense(1), where $m$ and $u$ are chosen based on the specific test task and can vary from task to task.

---

[6]For example, in one of the five folds, we trained the models on $4 \times 9$ tasks derived from CIFAR-10, CIFAR-100, MNIST and Fashion MNIST datasets, and evaluated the model on the 9 tasks derived from the EMNIST dataset

- **Prototypical Net**: the embedding model is 2-`Dense`(32).

- **Relation Net**: the embedding model is 2-`Dense`(32), and the relation model is 2-`Dense`(16) → `Dense`(1).

- **DNN + MAML**: 4-`Dense`(64) → `Dense`(1), 1200 training batches of size 256 where each batch contains 10 tasks, inner learning rate is 0.01, outer learning rate is 0.005, fine-tune 6 epochs.

- **Joint DEN**: `PLF` with 10 calibration keypoints, the distribution embedding model is 3-`Dense`(64) → `BatchAvg`, the Deep Sets classification model is 3-`Dense`(64) → `PairAvg` → 3-`Dense`(64) → `Dense`(1).

- **Conditional DEN**: `PLF` with 10 calibration keypoints, the distribution embedding model is 4-`Dense`(16) → `BatchAvg`, the Deep Sets classification model is 4-`Dense`(64) → `PairAvg` → 4-`Dense`(64) → `Dense`(1).

Remark that, during hyperparameter tuning, the maximum number of ReLU-activated dense layers we have tried is 12 and the maximum number of units we have tried is 64 for all models. The proposed Joint DEN and Conditional DEN tend to prefer larger models than the rest ones.

## C APPENDIX: ADDITIONAL RESULTS FOR MODEL AGGREGATION EXPERIMENTS

In this section, we present results for Joint DEN and Conditional DEN with fine-tuning in model aggregation experiments. For comparison, we also report results for Direct DNN and DNN with MAML.

### C.1 FIXED NUMBER OF FEATURES

We consider a support set of size 500. For Joint DEN and Conditional DEN, we use cross-validation on the support set to tune the hyperparameters—tuning epochs, initial learning rate and batch size. For DNN with MAML, we also tune the number of tuning layers, *e.g.*, fine-tune the last one or two dense layers, in addition to the above three hyperparameters.

Table 6: Test % AUC (standard error) on aggregating 25 classifiers with 500 supporting examples. Bold fonts indicate the best method at 95% significance level.

| Method | Test AUC (%) | | | |
|---|---|---|---|---|
| | Task 1 | Task 2 | Task 3 | Task 4 |
| Direct DNN | 92.07 (0.038) | 90.48 (0.029) | 97.62 (0.012) | 97.76 (0.008) |
| DNN + MAML | 91.94 (0.040) | 90.22 (0.029) | 98.13 (0.008) | 98.13 (0.006) |
| Joint DEN | 92.19 (0.030) | 89.97 (0.022) | 96.89 (0.030) | 97.32 (0.004) |
| Conditional DEN | 92.26 (0.030) | 89.65 (0.022) | 97.58 (0.025) | 97.18 (0.004) |

### C.2 VARIABLE NUMBER OF FEATURES

We repeat the procedures in the above section for the case when the number of features can vary.

## D APPENDIX: DESCRIPTION OF REAL DATASETS

We use three datasets in our real data analysis.

With Diamonds data, we use numeric carat, cut in ordinal scale, color in ordinal scale, clarity in ordinal scale, numeric depth and numeric table to classify whether the price of diamond is above \$951, above \$2402, or above \$5325, which corresponds to the 25% qunatile, median and 75% quantile in the data. Four of the six features, caret, cut, color and clarity are expected to be monotonic, whereas depth and table have unknown monotonicity.

Table 7: Test % AUC (standard error) on aggregating variable number of classifiers with 500 supporting examples.

| Method | Test AUC (%) | | | |
|---|---|---|---|---|
| | Task 1 | Task 2 | Task 3 | Task 4 |
| Direct DNN | 90.79 (0.137) | 89.87 (0.063) | 97.92 (0.021) | 97.66 (0.019) |
| DNN + MAML | 89.63 (0.130) | 89.51 (0.044) | 97.92 (0.031) | 97.72 (0.037) |
| Joint DEN | 91.23 (0.077) | 88.75 (0.031) | 96.57 (0.010) | 96.76 (0.007) |
| Conditional DEN | 91.74 (0.079) | 89.88 (0.029) | 97.82 (0.038) | 97.50 (0.007) |

With Nomao data, we use fax trigram similarity score, street number trigram similarity score, phone trigram similarity score, clean name trigram similarity score, geocoder input address trigram similarity score, coordinates longitude trigram similarity score and coordinates latitude trigram similarity score to classify whether two businesses are identical. All of the features are outputs from some other models, and are expected to be monotonic. Six and seven of the features have missing values. Fax trigram similarity score is missing 97% of the time; phone trigram similarity score is missing 58% of the time; street number trigram similarity score is missing 35% of the time; geocoder input address trigram similarity score is missing 0.1% of the time, and both coordinates longitude trigram similarity score and coordinates latitude trigram similarity score are missing 55% of the time.

With Puzzles data, we use has photo (whether the reviews has a photo), is amazon (whether the reviews were on Amazon), number of times users found the reviews to be helpful, total number of reviews, age of the reviews and number of words in the reviews to classify whether the units sold for the respective puzzle is above 24, above 45 or above 93. Most of the features have unknown monotonicity in their effect in the label.

