# OpenReview forum: "Distribution Embedding Network for Meta-Learning with Variable-Length Input"
_ICLR.cc/2021/Conference — Reject_

### Official Review · AnonReviewer3 · 2020-10-26
**Novel method; unsure about evalution / impact**

**Rating:** 5
**Confidence:** 2

**Review:**

Summary:
The authors proposed a method for meta-learning that produces a distributional embedding of the task, and then uses this information to perform few-shot classification. The method decouples feature extraction from feature aggregation & few-shot reasoning within the given task. The authors conduct experiments to show that their method is competitive with a few other methods from the literature on a number of datasets.


Strengths:
* The proposed method is novel.
* Based on the provided experiments, it appears to perform well.

Weaknesses:
* The motivation for this method is not very well explained. The authors discuss existing families of meta-learning methods & how their method does not fit into any of these, but it is not clearly explained what advantages theirs offers or where existing methods are lacking. Does the proposed method have any inductive biases or other desirable properties?
* The experiments are hard to interpret. Prior work on meta-learning has introduced a few standardized datasets (for example, Omniglot, mini-Imagenet, and tiered-Imagenet), and there is a large body of work that has relied on these datasets for evaluation. It would be helpful to benchmark the proposed method on those datasets so that it can be more thoroughly compared to existing work, helping better assess the impact of this method.
* Reproducibility / experimental details: Overall, I feel that it would be difficult to reproduce this work based on the content of the paper -- do the authors have plans to release the code? Several nontrivial details seem to be omitted -- for example, where do the features that are input to the proposed method come from? Is there any learned embedding / what happens on high dimensional problems?

Overall, I would lean toward rejection based on my current understanding of this paper, but I would be willing to revise my score based on the authors' response & the other reviews.

---

### Official Review · AnonReviewer2 · 2020-10-27
**Incomplete architecture analysis**

**Rating:** 4
**Confidence:** 4

**Review:**

This paper tackles the problem of meta-learning, namely learning from labelled datasets across related tasks, aiming for adaptation to unseen tasks, with little data. The focus is made on varying distributions across data features as well as varying numbers of features.

The proposed architecture (DEN) contains three building blocks:
1.	A transformation layer modeled as a piecewise linear function, that balances data distributions across tasks
2.	A distribution embedding module that can take variable-length inputs by considering subsets of features, either in a conditional form or a joint one
3.	A classification module that aggregates the input features and the distribution embedding, taking the form of a Deep Sets (DS) network.

On the applicative side, a novel method for task generation is leveraged using binary classifiers. The proposed network is applied to a classifier aggregation task, as well as seven target tasks from real datasets.

Strong points of the paper include:
1.	The writing of the paper is very clear overall.
2.	The paper tackles an important topic, and its focus points are valuable to the literature.
3.	The experiments procedure presented in section 5.1 and 5.2 is rigorous.

Some important explanations on modelization choices are missing from the current version of the paper, including the following ones:
1.	Authors do not mention why the set representation and related topology are adapted to pairs of features and labels.
2.	Authors resort to the DS framework; however, they do not motivate the use of invariant architectures for meta-learning to begin with, and the literature review on invariant networks is missing.
3.	The invariance actually designed here is particularly unclear. I do not think the invariance the authors want to consider is h being permutation invariant in Equation 6, as is hinted at. Invariances should also be investigated in the other steps, for instance Eq. 4, and labels should be particularized.
4.	In section 5.3, the fact that missing values are not handled similarly in DEN and in its competitors is not justified. Could you please clarify?

I recommend a reject for this paper, on the grounds that the architecture design needs further analysis (as explained above).

Minor details:
1.	In Section 4.2, maybe recalling the definition of sufficient statistics would clarify lemma 2.
2.	Typo in Eq 4: r instead of M.
3.	The notation g’ in Eq 5 could be misleading.

---

### Official Review · AnonReviewer1 · 2020-10-28

**Rating:** 4
**Confidence:** 3

**Review:**

This work proposes Distribution Embedding Network (DEN), a meta-learning model for classification. DEN is designed for the setting where data distribution and the number of features can vary across tasks. It classifies examples based on an embedding of data distribution.

The network first embeds data X using a piecewise linear function (PLF). They construct distribution embeddings by aggregating all subsets of size r and inputting to a Deep Sets classifier that takes support set distribution embeddings and query datapoints.

On page 5, the paper says that each task gets its own PLF. How is the PLF trained on meta-test tasks? Do you temporarily split the support set into train and validation sets to minimize the loss? This is a critical detail and should be clearer.

I think this method is more accurately described as a member of the neural process (NP) family. The overall structure is very similar to the conditional NP (CNP) model for classification [1]. Compared to CNP, the novel components are the PLF and their specific r-subset aggregation scheme (4, 5). However, the experiments do not sufficiently show how much each of these components contributes to performance. Rather than ProtoNets, RelationNets, MAML, etc., comparing against more relevant baselines such as PLF+CNP, r-subset + CNP, CNP+finetuning, etc. would have more clearly shown the efficacy of DEN.

Furthermore, many works confirm that using attention instead of mean-pooling increases performance [2,3,4]. Since attention plays a similar role (encoding high-order interactions among items) to the proposed r-subset aggregation scheme, comparing these two aggregation methods would more firmly ground DEN in the context of existing work.

[1] Garnelo, Marta, et al. "Conditional neural processes."

[2] Kim, Hyunjik, et al. "Attentive neural processes."

[3] Lee, Juho, et al. "Set transformer: A framework for attention-based permutation-invariant neural networks."

[4] Le, Tuan Anh, et al. "Empirical evaluation of neural process objectives."

minor

-The proof of Lemma 1 defines a new loss, which is an expectation of the loss over the data distribution. I feel that both the claim and proof are obvious.

-Section 4.1, just above (2): y_T \in R^n → y_T \in {0, 1}^n

-Figure 1: link boxes in odd places.

---

### Official Review · AnonReviewer4 · 2020-10-28
**Review of Distributed Embedding Network for Meta-learning with Variable-Length Input**

**Rating:** 4
**Confidence:** 3

**Review:**

############################################

Summary

This paper provides an interesting model that can be used for a meta-learning situation where both data distribution and the number of features vary across tasks. The proposed model referred to as ‘Distributed Embedding Network (DEN)’ transforms features to belong to the same distribution family, learn distribution embedding with them and process variable-length inputs by using DeepSets. Through experiments on binary classification tasks, this paper shows applying the proposed method on the problems outperforming meta-learning baselines.

############################################

Reasons for score

I vote for rejecting. This paper proposes a novel method that can handle varying data distribution and variable-length input for the meta-learning field. My main concerns are the clarity of the paper including the objective of each module design and that experiments are not enough to demonstrate the effectiveness of the proposed method. I hope the authors address my concerns during the rebuttal period.

############################################

Strong Points

1. This paper proposes a novel method that encodes varying data distribution across tasks for the meta-learning field.
2. Also, this paper can handle variable-length input across tasks.
3. This paper demonstrates their potential to be developed by outperforming baselines on binary classification tasks.

############################################

Weak Points

1. The paper said 'we compare ~ with a range of state-of-the-art baseline models' but, as I know, PrototypicalNet (2017), RelationNet(2018), MAML (2017) are not the SOTA baseline models. Also I think the baseline models used in the experiments are not enough strong. Could the authors show more results of Table 1 or 2 performed by recent meta-learning methods?
2. I feel the meaning of the same parametric family’ is rather ambiguous. Could the author describe the definition of ‘same parametric family’?
3. Additionally, on the 3 page of the paper,  'new unseen tasks whose data distribution falls in the same parametric family.' is written. but, I think if unseen tasks such as tasks sampled from out-of-distribution are enough different from training tasks, it is difficult to use the shared parameters which are trained by training seen tasks.
4. I wonder why the model with equation (2) can learn distribution. Please give me a more description including the role of pair-wise product between features in the equation (2).
5. In the section 4.1.2, the reason decomposing the joint dist. into smaller pieces to handle variable-length features is unclear for me. Could the author give a more explanation for it?
6. Even the paper said 'extending to multi-class classification is trivial', but I hope the experiment results on multi-class classification are included in the paper. Could the author show experiments on multi-class classification problems (e.g. 5way, 10way)?
7. I think that the overall writing of the paper should be improved.

    (1) I feel the introduction section is rather lacking in abstract level explanation of the proposed method. Adding a concept figure is one of options for a better explanation.

    (2) Overall notations are rather complicated for me. Also, some notation looks written wrongly. Notation shapes of z of (5) and the sentence above (5) are different. Are they have different meanings? How about in case z of (4) and (5)?

    (3) Before using DeepSet, it would be better to add a description of DeepSet.

    (4) is \psi in (6) DeepSet?
8. The name ‘DEN’ is already used in the paper ‘Lifelong Learning with Dynamically Expandable Networks (ICLR2018)’[1]. I recommend this paper change the abbreviation.
9. SetTransformer[2] is an advanced set encoding module outperforming DeepSet. I recommend using SetTransformer instead of DeepSet in the future.

[1] Lifelong Learning with Dynamically Expandable Networks, ICLR2018

[2] Set Transformer: A Framework for Attention-based Permutation-Invariant Neural Networks, ICL2019

---

### Author Response · Authors · 2020-11-24
**Responses to Reviewers' Comments**

We thank all reviewers for their insightful and concrete comments. We address their main comments below.

R1. How is the PLF trained on meta-test tasks?

We directly train the PLF on the support set without splitting it into training and validation sets. Note that we did not tune hyperparameters in this step so there is no need for a validation set.

Comparing against more relevant baselines such as PLF+CNP and r-subset + CNP would have more clearly shown the efficacy of DEN.

We thank the reviewer for pointing out the connection of our paper with CNP. We have added discussion and citation of CNP in the paper. We agree that our method could belong to the CNP family, although CNP itself is not designed for variable-length features. Thus, special treatments are needed, such as PLF and r-subset. Comparing our method against (modified) CNP is out of the scope of this paper. We leave it to future work.

Since attention plays a similar role to the proposed r-subset aggregation scheme, comparing these two aggregation methods would more firmly ground DEN in the context of existing work.

The Deep Sets structure is just a component of our proposal and not the main part. One can freely choose other aggregation schemes, including set transformers. Comparing against set transformers is not the goal of this paper. We have made it clear in the Conclusion section of the revised paper.

R2. We reply to the points under “Some important explanations …” below.

1. The reason we use pairs of features is as follows. Let us consider a single feature vector $x = (x_1, \dots, x_d)$ for simplicity. If $x_1, \dots, x_d$ are mutually independent, then the joint distribution $p(x)$ factorizes as $p(x_1) \dots p(x_d)$. In this case, we can take the features $x_i$ from a batch of examples to estimate the marginal $p(x_i)$. However, this independence assumption is not realistic, so we use pairwise distributions $p(x_i, x_j)$ (or r-tuples in general) to estimate the joint $p(x)$. In fact, this is sufficient in the Gaussian example we considered.

2. We agree with the reviewer that the discussion on invariant networks is insufficient. We have  added references in the Conclusion section of the revised paper.

3. We thank the reviewer for catching this point. Indeed, we did not mean to require h to be permutation invariant. What we want is the whole model, after the PLF layer, is permutation invariant to the transformed feature z.

4. For competing methods, there is not a clear optimal strategy to handle missing values similar to the one in DEN since they do not contain a PLF layer. What we did is common in practice, and, in fact, more “flexible” than the PLF strategy in our proposal. We also have other experiments that do not have missing values, where DEN also outperforms competing methods.

R3. We reply to the points under “Weaknesses” below.

1. We have a discussion in the second paragraph in Sec. 1. The current meta-learning methods are limited to tasks that are of similar feature structure, e.g., image classifications tasks, which impose 1) constraints on its applications and 2) challenges in collecting pretraining tasks. Our proposal can be applied to tasks with features of variable length, allowing us to 1) apply the model pretrained on image classification tasks to completely different tasks like the ones we considered in Sec. 5.3 and 2) massively simulate pretraining tasks using the methodology proposed in the paper.

2. These standardized datasets are all for multi-class classification tasks. In this paper we consider the applications when the distribution and number of features are different across tasks. Such applications are much more prevalent in binary classification tasks than multiclass classification tasks.

3. Yes, we have indicated that in the footnote on page 5.

R4.

1. We thank the reviewer for this question. We have modified the paper to make it clear that those three methods are representative methods in their respective camp, but are not SOTA.

2. We call a set of densities $\{f(\cdot, \theta): f: R^d \times \Theta \to R, \theta \in \Theta\}$ a parametric family.

3. We have a discussion on this in the second paragraph on page 3.

4. This is because the sufficient statistics of Gaussian distribution are its first and second moments.

5. Since every basic function must have a fixed number of arguments, we have to determine a fixed size in order to handle features of variable length. What we did is to decompose the feature vector into subvectors of a fixed length, then we can apply a function to each of these subvectors. This is just one possible strategy, but it is sufficient in the Gaussian case.

6. In classical ML applications with variable length features, binary classification is much more common than multiclass classification. Moreover, since our idea is new, we want to focus on binary classification tasks to illustrate it clearly. We will consider multi-class classification tasks in future work.

---

### Decision · Program_Chairs · 2021-01-07
**Final Decision**

**Decision:**

Reject

**Comment:**

This paper addresses a meta-learning method which works for cases where both the distribution and the number of features may vary across tasks. The method is referred to as 'distribution embedding network (DEN)' which consists of three building block. While the method seems to be interesting and contains some new ideas, all of reviewers agree that the description for each module in the model is not clear and the architecture design needs further analysis. In addition, experiments are not sufficient to justify the method. Without positive feedback from any of reviewers, I do not have choice but to suggest rejection.